# A Trade-Off between Complexity and Interaction Quality for Upper Limb Exoskeleton Interfaces

**DOI:** 10.3390/s23084122

**Published:** 2023-04-20

**Authors:** Dorian Verdel, Guillaume Sahm, Olivier Bruneau, Bastien Berret, Nicolas Vignais

**Affiliations:** 1Université Paris-Saclay, CIAMS, 91405 Orsay, France; 2CIAMS, Université d’Orléans, 45100 Orléans, France; 3LURPA, ENS Paris-Saclay, Université Paris-Saclay, 91190 Gif-sur-Yvette, France

**Keywords:** human-exoskeleton interactions, self-aligning mechanisms, passive degrees of freedom

## Abstract

Exoskeletons are among the most promising devices dedicated to assisting human movement during reeducation protocols and preventing musculoskeletal disorders at work. However, their potential is currently limited, partially because of a fundamental contradiction impacting their design. Indeed, increasing the interaction quality often requires the inclusion of passive degrees of freedom in the design of human-exoskeleton interfaces, which increases the exoskeleton’s inertia and complexity. Thus, its control also becomes more complex, and unwanted interaction efforts can become important. In the present paper, we investigate the influence of two passive rotations in the forearm interface on sagittal plane reaching movements while keeping the arm interface unchanged (i.e., without passive degrees of freedom). Such a proposal represents a possible compromise between conflicting design constraints. The in-depth investigations carried out here in terms of interaction efforts, kinematics, electromyographic signals, and subjective feedback of participants all underscored the benefits of such a design. Therefore, the proposed compromise appears to be suitable for rehabilitation sessions, specific tasks at work, and future investigations into human movement using exoskeletons.

## 1. Introduction

Active exoskeletons are considered to be potentially very beneficial for the future of rehabilitation protocols [1,2,3,4] and work-related musculoskeletal disorder (MSD) prevention [5,6,7,8]. A well-known limitation of such systems is the unavoidable joint misalignment (JM) between the human and the exoskeleton [9]. This leads to a kinematic incompatibility, also called hyperstaticity, as supported by both geometric models and the theory of mechanism-based models [10,11,12,13]. Such kinematic incompatibility can result in an increase in human-exoskeleton interaction efforts [12,14,15] and significant discomfort [15]. Hence, designing exoskeletons that reduce or compensate for this kinematic incompatibility is a key factor for their future widespread implementation.

Two main categories of solutions have been identified in the literature to solve these concerns [16]. In the first category, the solution consists of designing non-anthropomorphic exoskeletons. In the second category, the solution consists of compensating the JM using adjustable anthropomorphic exoskeletons and passive degrees of freedom at the human-exoskeleton interfaces level. On the one hand, non-anthropomorphic designs refer to exoskeletons that violate their most widespread definition [17]: “an exoskeleton is an external structural mechanism with joints and links corresponding to those of the human body”. For example, recent works based on a modular synthesis approach have led to the introduction of two four-bar mechanisms in series to design an upper limb exoskeleton [18,19]. Overall, numerous designs have been proposed for the main joints of both the upper [20,21,22] and lower [23,24,25,26,27] human limbs. On the other hand, anthropomorphic designs comply with the previous definition and must incorporate JM compensation methods. From that perspective, the first possibility is to design exoskeletons with adjustable limbs [28,29,30,31,32]. However, such features must be incorporated from the beginning of the design process because they determine the mechanical structure of the exoskeleton. A second possibility, which is the solution adopted in the present paper, is to design self-aligning human-exoskeleton interfaces. Such interfaces have been successfully added to upper-limb exoskeletons to improve the interaction quality [12,13,15]. The same positive results were obtained for lower-limb exoskeletons (see [33] for an example on the knee joint).

Importantly, all of these solutions are known to increase the exoskeleton’s mechanical complexity and inertia, which can result in unwanted interaction efforts and a complex controller design [9,14,16]. In the present paper, we thoroughly investigate the influence of two passive rotations included in the forearm interface of an upper-limb exoskeleton during shoulder and elbow reaching movements performed in a sagittal plane. Such a design has been shown to be highly beneficial during elbow movements performed in a sagittal plane with a transparent exoskeleton [15]. In such a framework, interaction efforts are minimized [34,35,36] to properly isolate the effects of the tested interfaces. In the case of 2-degrees of freedom reaching movements in a sagittal plane, this design constitutes a new trade-off between complexity and interaction quality. Indeed, the absence of passive joints in the arm interface allows for limiting the added inertia and complexity [37]. Furthermore, the passive rotations included in the forearm interface could have global and beneficial effects for both human effort and comfort, thereby limiting the alterations of natural human movements. Therefore, the aim of this study is to analyze the effects of two passive rotations included in the forearm interface of an upper-limb exoskeleton on the quality of the interaction during reaching movements performed in a sagittal plane.

The remainder of the present paper is organized as follows: in Section 2, the materials and measurements, the exoskeleton control, and the evaluation task implemented to test the effects of the different interfaces are described in detail. Furthermore, in Section 3, the evaluation results are described in terms of interaction efforts, kinematics, and EMG signals to provide an overview of the effects of the different interfaces. Finally, these results are discussed in Section 4.

## 2. Materials and Methods

### 2.1. Participants and Materials

#### 2.1.1. Participants

A total of N=19 right-handed, healthy young adults (7 females) were involved in the experiment. The anthropometric characteristics are the following: age 24 ± 2 years old, weight 66.7±11 kg, and height 1.73±0.07 m. All participants provided their written informed consent before performing the experiment, as required by the Helsinki Declaration.

#### 2.1.2. Kinematic Recordings

Human movement kinematics were recorded by means of an optoelectronic system (10 Oqus 500+ cameras, sample rate 179Hz; Qualisys, Gothenburg, Sweden). Eight 10mm reflective markers were placed on the participants to track their movements and provide a satisfying convergence of the labeling algorithm (Qualisys Track Manager, Qualisys, Gothenburg, Sweden): solar plexus, sternoclavicular joint, acromion, epicondyle, and epitrochlea of the elbow, middle of the forearm, styloid process of the radius (due to the human-exoskeleton configuration, the styloid process of the ulna was not accessible) and the base of the index finger. A 3mm reflective marker was placed at the tip of the participant’s index. We used the recorded 3-dimensional positions of this last marker to describe the human movement kinematics.

#### 2.1.3. EMG Recordings

The human muscle activity was quantified by means of surface EMG (Wave Plus wireless EMG system, sample rate 2kHz, Cometa, Bareggio, Italy). Six muscles of the forearm, arm, and shoulder, involved in sagittal plane reaching movements, were recorded: brachioradialis (elbow flexor), biceps brachii (elbow and shoulder flexor), triceps brachii lateral head (elbow extensor), long head (elbow and shoulder extensor), anterior deltoid (shoulder flexor), and posterior deltoid (shoulder extensor). Before placing the electrodes, the participants were locally shaved and a hydroalcoholic solution was applied. The electrode positioning followed the SENIAM recommendations [38].

#### 2.1.4. Ergonomic Feedback Questionnaire

The feedback of participants, with regard to the different conditions performed with the exoskeleton, was collected through the following semi-directed questionnaire, including negative and positive questions, inspired by previous studies [15,39,40]:Comfort:
-Did you feel any friction or irritation during the movement? (negative)-Did you experience any pressure points at the level of the interfaces? (negative)-Rate the general comfort of movement with the current interfaces. (positive)Movement ability:
-Rate the mobility with the current interfaces. (positive)-Did you feel any constraint on your motion range? (negative)Accuracy:
-Was it easy to reach the targets? (positive)

For each of these six items, participants gave a grade between 1 (if they completely agreed with a negative item or completely disagreed with a positive item) and 5 (if they completely agreed with a positive item or completely disagreed with a negative item). The grading scale proposed to participants was as follows: 1 and 5, completely agree or disagree; 2 and 4, agree or disagree; and 3, neither agree nor disagree.

#### 2.1.5. ABLE Exoskeleton and Tested Interfaces

The present experiments were conducted using an ABLE active upper limb exoskeleton, known for its transmission system that provides high back-drivability and potential transparency [29,41,42]. This exoskeleton includes three active degrees of freedom reproducing the main rotations of the shoulder (i.e., internal/external, abduction/adduction, and flexion/extension), and one active degree of freedom reproducing the main rotation of the elbow (i.e., flexion/extension). The human was connected to the exoskeleton at the level of the forearm and the level of the arm (see Figure 1). The exoskeleton was modified to include two 6-axe force/torque (FT) sensors (1010 digital FT, ATI, sample rate 1 kHz) placed at the level of the human-exoskeleton interfaces to quantify and minimize (through the exoskeleton control) the interaction efforts.

Previous works conducted on this exoskeleton have led to the inclusion of several passive degrees of freedom at the level of the forearm interface to improve the quality of the interaction during elbow flexion/extension in a sagittal plane [15]. In particular, a passive translation along the forearm of the exoskeleton (i.e., $x_{\mathrm{FA}}$, see Figure 1A) and two passive rotations preventing the apparition of flexion torques during elbow movements (i.e., around $y_{\mathrm{FA}}$ and $z_{\mathrm{FA}}$, see Figure 1A) were included. Furthermore, two thermoformed orthoses were designed to increase the human-exoskeleton interaction area both at the forearm and arm levels, which has been suggested to improve the comfort reported by users [15]. The forearm orthosis was designed to block the human wrist rotations.

In the present paper, we tested the effect of these two passive rotations during reaching movements performed in the sagittal plane, where the degree of hyperstaticity was modified by the arm interface (which does not include passive degrees of freedom). The first two active rotations of the exoskeleton (i.e., shoulder internal/external rotation and abduction/adduction) were mechanically blocked using dedicated devices to ensure that movements were performed in a sagittal plane. Consequently, two conditions were tested in the current experiment: a condition without passive degrees of freedom at the forearm level (*noRot*) and a condition with these two degrees of freedom (*Rot*). The rotations were mechanically blocked using dedicated screws during the *noRot* condition.

### 2.2. Evaluation Task

The task was composed of reaching movements in a sagittal plane toward two semi-spherical 3cm targets. Reaching tasks in a sagittal plane have been widely studied in the rehabilitation literature, and exoskeletons may be of potential help for this kind of application [43,44,45,46]. Before movement onset, participants were asked to remain static in a reference position, i.e., with the arm remaining vertical and the forearm horizontal. Then, one of the two targets was lit for 4±0.5s, which allowed the participant to perform the movement toward the target at a comfortable and self-selected pace without being able to anticipate. After this duration, the target was turned off and the participant was asked to return to the reference position. A total of 30 reaching movements were performed. Half of these movements were performed toward a target higher than the reference position (referred to as $T_{top}$, aligned $30^{{}^{\circ}}$ above the reference position, as in Figure 1B) and the other half toward a target lower than the reference position (referred to as $T_{bot}$, aligned $30^{{}^{\circ}}$ below the reference position). This allowed assessing possible different constraints applied by the exoskeleton at the extremities of the large workspace. The sequence of reaching movements was pseudo-random.

Three blocks were performed: one with each of the two tested configurations of the forearm’s human-exoskeleton interface (i.e., *noRot* and *Rot* conditions) and one outside the exoskeleton (referred to as *noExo*), which served as the baseline for comparisons. To avoid potential wrist rotations and remain comparable between conditions in terms of kinematics, participants wore a light splint during the *noExo* block.

The exoskeleton was controlled in a transparent mode, which means that interaction efforts were minimized to minimize the possible alterations of human movements [34,35]. The controller was based on the identification of the exoskeleton’s dynamics, performed following previous procedures [36], coupled to a force feedback minimization [47]. More precisely, the controller was structured as detailed in Figure 2, which corresponds to the following equation,
(1)$$\tau_{r}=-K_{p}Lf_{e}-K_{i}\int_{t_{0}}^{t}Lf_{e}\;dt+{\hat{\tau }}_{m}$$
where $\tau_{r}$ is a 2×1 vector representing the joint torques applied by the controller at the shoulder and elbow levels, respectively (the motor current is $I_{r}=K_{t}\tau_{r}$, where $K_{t}$ is the torque constant of the motors, which in the present case is a scalar because both motors are the same), $K_{p}$ and $K_{i}$ are 2×2 diagonal matrices corresponding to proportional and integral control gains, respectively, and L is a 2×2 diagonal matrix corresponding to the lever arms. The controlled forces are represented by $f_{e}={\left(F_{z,A},F_{z,\mathrm{FA}}\right)}^{⊤}$, which is the vector of normal forces applied to the human at the arm and forearm levels, respectively (see Figure 1A for the definition of the normal axes). Finally, the estimated motor torque ${\hat{\tau }}_{m}$, based on the identification of the exoskeleton dynamics [36], compensated for the gravity and friction torques of the exoskeleton.

### 2.3. Data Processing

#### 2.3.1. Kinematics

The successive positions of the index reflective marker were low-pass filtered (Butterworth, fifth order, 5Hz cut-off frequency) before numerical differentiation, which allowed for the computation of the velocity and acceleration profiles. Movement onset and end were defined with a threshold fixed at 5% of the peak velocity [36,48,49,50]. The movement duration (MD) and amplitude were defined as the elapsed time and the traveled distance between these bounds, respectively. The average velocity (aV) was computed as the ratio between the amplitude and MD the mean acceleration during the acceleration phase was computed as the ratio between the peak velocity and the time elapsed between the movement onset and the peak velocity. Such early acceleration parameters have been shown to be impacted by additional inertia and unwanted interaction efforts at the interface level [15,51].

#### 2.3.2. EMG

EMG data were band-pass filtered (Butterworth, fourth order, [20;450]Hz cut-off frequencies) before being centered and rectified [52]. The root mean square (RMS) of the signal was then computed between the movement onset and end, based on the kinematics segmentation. This parameter allowed quantifying the average effort provided by a participant over the movement period. The computed RMS was averaged according to the functional muscle groups previously defined (i.e., shoulder or elbow flexors/extensors, see Section 2.1.3). The RMS of one muscle was computed as follows,
(2)$$\mathrm{RMS}=\sqrt{\frac{1}{N}\sum_{i=1}^{N}{\mathrm{EMG}}_{i}^{2}}$$
where *N* is the number of samples and ${\mathrm{EMG}}_{i}$ is the *i*th sample.

#### 2.3.3. Interaction Efforts

Interaction efforts measured at the level of the human-exoskeleton interfaces were low-pass filtered (Butterworth, fifth order, 5Hz cut-off frequency). The absolute maximum and absolute average values recorded for each component were used as descriptors of the interaction efforts. The maximum value provided information regarding the risk of inducing pain at the interface level [15,53,54,55], while the average value provided information regarding the long-term acceptability of the device [34,53].

### 2.4. Statistical Analysis

The possible main effects of the tested condition (i.e., *noExo noRot* and *Rot*) on the interaction were first assessed by one-way repeated measures ANOVA. Whenever a potential sphericity issue was detected (i.e., ϵ<0.75) a Greenhouse–Geisser correction was applied. The significance level of the ANOVA was set at p<0.05. Whenever a significant difference was reported, the $\eta^{2}$ was provided to illustrate the effect size.

In the case of the main effect of the condition on a parameter, pairwise *t*-tests were performed between conditions for each target separately, which allowed assessing potential non-homogeneous effects across the workspace. The significance level of these comparisons was set at p<0.05. Whenever a significant difference was reported, Cohen’s *D* was provided to illustrate the effect size.

All statistical analyses were performed using custom Python 3.8 scripts and the Pingouin package [56].

## 3. Results

In this section, the effects of passive rotations included in the forearm human-exoskeleton interface are extensively assessed. First, the interaction efforts at the arm and forearm levels are described and compared between the *noRot* and *Rot* conditions (Section 3.1). Second, the impacts of both conditions on human kinematics are assessed by comparing these two conditions to each other and to natural human movements recorded in the *noExo* condition (Section 3.2). Third, the same types of comparisons are detailed on EMG signals to assess the level of human muscle solicitation (Section 3.3). Finally, the subjective reports of participants are detailed for both the *noRot* and *Rot* conditions (Section 3.4).

### 3.1. Effects on Interaction Efforts

The interaction efforts were assessed by computing the absolute maximum and absolute average values of each of the interaction forces and torque components. This analysis was performed for both the arm and forearm FT sensors and is summarized in Figure 3.

*Effects of the tested condition:* As reported in Figure 3A, the main effect of the condition was found for the z component of the arm interaction force. Despite the important effect size returned by the ANOVA ($\eta^{2}>0.2$), pairwise comparisons were not significant. To summarize, there was a significant reduction in the *Rot* condition for the *z* component of the arm interaction force, but not for the other components. Furthermore, the *Rot* condition did not induce any significant change in the arm interaction torques (see Figure 3C).

As reported in Figure 3B, the main effects of the condition were found for both the maximum and average values of the *y* and *z* components of the interaction forces. The pairwise comparisons performed for the *y* and *z* components highlighted a reduction of the interaction forces with the *Rot* condition, when compared to the *noRot* condition, for both targets (in all cases: p<0.011, D>0.87). The *Rot* condition induced overall lower interaction forces at the forearm level when compared to the *noRot* condition.

As reported in Figure 3D, the main effect of the condition was found on the *y* and *z* components of the interaction torques. The pairwise comparisons performed for the *y* and *z* components highlighted a reduction of the interaction torques with the *Rot* condition (when compared to the *noRot* condition) for both targets (in all cases: $p<3.10^{-4}$, D>1.32). To summarize, the *y* and *z* components of the interaction torques were clearly reduced in the *Rot* condition when compared to the *noRot* condition.

*Effects of the aimed target:* The ANOVAs performed on the arm interaction torques returned the main effects of the target in the *y* and *z* components. In particular, these components were significantly smaller for $T_{bot}$ than for $T_{top}$, both in terms of maximum and average values (in all cases: $F_{1,18}>10.16$, p<0.0051, $\eta^{2}>0.36$), which was confirmed by pairwise comparisons (in all cases: p<0.0055, D>0.65). To summarize, the arm interaction torques were higher when aiming at $T_{top}$ when compared to those reported when aiming at $T_{bot}$. Furthermore, there was no effect of the aimed target on the arm interaction force components.

The main effect of the target location was found in the *x* and *y* components of the forearm interaction forces, both in terms of maximum and average values (in all cases, i.e., $F_{1,18}>20$, $p<3.10^{-4}$, $\eta^{2}>0.52$). Subsequent pairwise comparisons showed that the *x* force component was higher for $T_{bot}$ than for $T_{top}$ both in terms of average and maximum values (in both cases: p<0.009, D>0.61). The target location modified the *x* component of the interaction force, with $T_{bot}$ inducing higher efforts than $T_{top}$ in this component. Furthermore, the target location did not have any significant effect on the forearm interaction torques.

### 3.2. Effects on Human Movement Kinematics

Kinematics were qualitatively assessed by means of their temporal evolution, as summarized in Figure 4. Qualitatively, the average trajectories provided in Figure 4A mainly tended to be shorter for the $T_{bot}$ target in the *noRot* condition when compared to the trajectories performed in the *noExo* and *Rot* conditions. Furthermore, the velocity profiles were, overall, bell-shaped for all conditions but the velocities tended to be lower in the *noRot* and *Rot* conditions when compared to the *noExo* condition (see Figure 4B), which was also true for the accelerations (see Figure 4C). Finally, the acceleration profiles reported in the *noRot* condition were noisy, particularly for movements toward $T_{bot}$.

To quantify human movement kinematics alterations, several descriptors were computed. They are summarized in Table 1.

*Movement duration:* (MD) The conducted pairwise comparisons exhibited an overall significantly higher MD in the *noRot* and *Rot* conditions than in the *noExo* condition (in both cases: p<0.006, D>0.64). Target-based comparisons revealed a higher MD in the *Rot* condition than in the *noExo* condition for $T_{top}$ (p=0.04, D=0.69) and a higher MD in the *noRot* condition than in the *noExo* condition for $T_{bot}$ (p=0.02, D=0.82). To summarize, both the *noRot* and *Rot* conditions induced higher MD than the *noExo* condition, the *noRot* condition was worse for $T_{bot}$ and the *Rot* condition was slightly worse for $T_{top}$.

*Amplitude:* The conducted pairwise comparisons highlighted that the movement amplitude was smaller in the *noRot* condition than in the *noExo* condition for $T_{bot}$ (p=0.016, D=1.11), which was the only significant effect. To summarize, the *noRot* condition impacted movement amplitude significantly and non-homogeneously (with regard to the workspace) when compared to the *noExo* condition.

*Average velocity* (aV): The conducted pairwise comparisons exhibited an overall lower aV in the *noRot* and *Rot* conditions than in the *noExo* condition (in both cases: p<0.003, D>0.7). Target-based comparisons revealed a lower aV in the *noRot* and *Rot* conditions than in the *noExo* condition for $T_{bot}$ (in both cases: p<0.016, D>0.81). To summarize, both conditions performed inside the exoskeleton tended to reduce aV; the effect was more pronounced for $T_{bot}$ than for $T_{top}$. Finally, the *noRot* condition induced a higher impact on aV than the *Rot* condition for $T_{bot}$.

*Mean acceleration* (mA): The conducted pairwise comparisons exhibited an overall lower mA in the *noRot* and *Rot* conditions than in the *noExo* condition (in both cases: p<0.003, D>0.69). Target-based comparisons revealed a lower mA in the *noRot* and *Rot* conditions than in the *noExo* condition for $T_{bot}$ (in both cases: p<0.04, D>0.68) and a lower mA in the *Rot* condition than in the *noExo* condition for $T_{top}$ (p=0.043, D=0.68). To summarize, both conditions performed inside the exoskeleton tended to reduce mA; the effect was more pronounced for $T_{bot}$ than for $T_{top}$. Finally, the *noRot* condition induced a higher impact on mA than the *Rot* condition for $T_{bot}$, with the opposite for $T_{top}$.

### 3.3. Effects on Human Muscle Activities

We quantified the average muscle effort of participants in each of the tested conditions by computing the average RMS of each of the functional muscle groups previously defined: shoulder flexors/extensors and elbow flexors/extensors (see Section 2.1.3 for details regarding the group composition). The results of these analyses are summarized in Figure 5.

*Shoulder flexors:* The conducted pairwise comparisons exhibited an overall higher RMS in the *noRot* condition than in the *noExo* and *Rot* conditions (in both cases: $p<2.10^{-4}$, D>0.99). Target-based comparisons revealed a higher RMS in the *noRot* condition than in the *noExo* and *Rot* conditions for $T_{top}$ (in both cases: p<0.01, D>0.87). The *noRot* condition also induced a higher RMS of shoulder flexors than the *noExo* and *Rot* conditions for $T_{bot}$ (in both cases: p<0.002, D>1.12). To summarize, the *noRot* condition clearly induced higher levels of shoulder flexor activity when compared to the *noExo* and *Rot* conditions, as verified independently of the target. Furthermore, the *Rot* condition did not induce significantly higher levels of shoulder flexor activity than the *noExo* condition.

*Shoulder extensors:* No main effect of the tested condition was found on shoulder extensors.

*Elbow flexors:* The conducted pairwise comparisons exhibited an overall higher RMS in the *noRot* condition than in the *noExo* and *Rot* conditions (in both cases: $p<7.10^{-6}$, D>1.11). Target-based comparisons revealed a higher RMS in the *noRot* condition than in the *noExo* and *Rot* conditions for $T_{top}$ (in both cases: $p<4.10^{-3}$, D>1.01). The *noRot* condition also induced a higher RMS of elbow flexors than the *noExo* and *Rot* conditions for $T_{bot}$ (in both cases: $p<7.10^{-4}$, D>1.2). To summarize, the *noRot* condition induced higher levels of elbow flexor activity when compared to the *noExo* and *Rot* conditions, which was verified for both targets. Furthermore, the *Rot* condition did not induce significantly higher levels of elbow flexor activity than the *noExo* condition.

*Elbow extensors:* The conducted pairwise comparisons exhibited an overall higher RMS in the *noRot* condition than in the *noExo* and *Rot* conditions (in both cases: p<0.033, D>0.49). To summarize, the *noRot* condition induced higher levels of elbow extensor activity than the *noExo* and *Rot* conditions.

### 3.4. Ergonomic Feedback

The questionnaire submitted to participants was designed to evaluate the overall comfort, movement ability, and accuracy through six questions (see Section 2.1.4 for the detailed questions). The results of this evaluation are summarized in Figure 6.

Overall, the grades given by participants were higher for the *Rot* condition than for the *noRot* condition, which suggested a better acceptability of the device when including passive rotations. Pairwise comparisons were performed for each of the subjective criterion to confirm or infirm the observed trends. These pairwise comparisons confirmed that the *Rot* condition induced lower friction/irritation (p=0.01, D=0.94), lower pressure (p=0.043, D=0.72), higher overall comfort (p=0.004, D=1.06), higher mobility ($p=10^{-5}$, D=1.8), and a higher motion range ($p=2.10^{-6}$, D=1.99) when compared to the *noRot* condition. Finally, the participants did not report a significant difference regarding their subjective movement accuracy between the *noRot* and *Rot* conditions.

## 4. Discussion

In the present paper, we tested the influence of two passive rotations included at the forearm interface on the quality of human-exoskeleton interactions during reaching movements in a sagittal plane. Although the inclusion of passive degrees of freedom has been shown to be an important factor to improve such interactions [12,15], the design of human-exoskeleton physical interfaces has received less attention than other developments, such as control methods or transmission designs. However, it has been previously shown that including such passive degrees of freedom in the exoskeleton design increases its inertia and mechanical complexity [9]. Hence, the robot control becomes more complicated, and unwanted interaction forces, due to an incomplete compensation of the dynamics (in particular, of the inertia), can be more important. Here, we tested whether a forearm interface including several passive degrees of freedom could be an acceptable compromise between improving the interaction quality and providing a simple mechanical design with limited additional inertia (i.e., limited due to the absence of passive degrees of freedom in the arm interface). Following previous evaluation methodologies [15,34], the human-exoskeleton interaction was quantified through a set of objective (i.e., interaction forces, kinematics, and EMGs), and subjective (i.e., questionnaires) metrics.

First, an overall reduction of the interaction forces and torques was observed with the *Rot* condition, as in other works [12,15]. In particular, the inclusion of two passive rotations drastically decreased both the interaction forces and torques at the level of the forearm interface. Interestingly, this reduction of interaction forces, in particular along $y_{\mathrm{FA}}$, which is orthogonal to the movement plane (i.e., it cannot be a motive in this task), suggests a reduction in the mobility constraints applied to the participants. However, the corresponding reduction was small at the level of the arm interface, which does not include passive degrees of freedom. Finally, the last important observation was the non-homogeneous magnitude of interaction efforts across the tested workspace. Indeed, overall, the interaction torques were higher during movements aimed at $T_{top}$ when compared to those aimed at $T_{bot}$ at the level of the arm interface, thereby exhibiting different constraints depending on the human pose. On the contrary, the interaction force along $x_{\mathrm{FA}}$, at the level of the forearm interface, was higher during movements aimed at the lower target $T_{bot}$ than during those aimed at $T_{top}$. This interaction force along $x_{\mathrm{FA}}$ was not modified by the passive rotations, mainly due to the weight of the sliding FT sensor embedded in the interface and to the weight of the passive slider joint, which generate different interaction efforts depending on the arm and forearm positions.

Second, the natural human kinematics were overall less impacted when including the passive rotations at the forearm level. Interestingly, the substantial reduction of the interaction efforts, which one might have expected to be reflected in the kinematics, was not as evident in these metrics. Overall, the reduction in aV was lower than 30%, which is a common value reported in previous studies using an open-loop compensation of the exoskeleton’s weight [29,34,51]. This was probably due to the closed-loop interaction minimization efforts. Furthermore, the main differences between the *noRot* and *Rot* conditions were for the lower target $T_{bot}$. Hence, as for interaction efforts, human kinematics are differently affected depending on the target location in the workspace. These observations are consistent with joint misalignment (JM) being essentially a geometric problem, as predicted by the theory of mechanisms [12,13] and more specific geometric models [10,11].

Third, the muscle activity levels during movement (described by the RMS of the EMG signals) were higher overall in the *noRot* condition than in the *Rot* and *noExo* conditions. Moreover, the muscle efforts were comparable between the *Rot* and *noExo* conditions. Primarily, this result is strongly in favor of the introduction of passive degrees of freedom at the forearm interface as an acceptable compromise because it does not increase substantially the level of muscle effort in the task, whether it be for the elbow or shoulder muscles. Furthermore, it is an interesting observation when coupled with the limited effects reported on kinematics because of their potential implications. Indeed, recent works have shown that humans are prone to expending large amounts of energy to save time [57], which was also shown while interacting with an exoskeleton [58]. Such an energetic expenditure was as predicted by the minimum time-effort theory in the field of human motor control [59,60], which is probably due to a temporal discounting of the reward (here, achieving the task) associated with the movement [61,62]. Hence, it is possible that in the present work, as previously suggested during the evaluation of the transparency of the ABLE exoskeleton [51], participants compensated for a part of the interaction efforts to save time and reproduce acceptable trajectories in the task space by expending energy. Interestingly, the level of comfort has been pointed out as a factor influencing humans’ preferred MD [63]. However, in the present study, it seems that the level of comfort was sufficient to allow participants to save time by expending more energy. Indeed, they could also have chosen to minimize their energetic expenditure by moving slower in the *noRot* condition than in the *Rot* condition, which was not clearly the case.

In the present experiment, participants reported an overall higher level of comfort and lower movement constraints with the *Rot* condition compared to the *noRot* condition. Interestingly, the average grades attributed to the *noRot* condition by the participants were around 3, which corresponds to “neither agree nor disagree”, in terms of general comfort and mobility. Such grades (and it is, thus, equally true for those attributed to the *Rot* condition) seem to be compatible with the application of higher levels of effort to save time. However, such a strategy might become detrimental to the skin at the level of the interface [55] and to the acceptability during long-term utilization of the device. Finally, it is worth mentioning that no effect of the condition was found on subjective accuracy, which was always well-graded. Consequently, the effects reported on movement vigor (i.e., self-selected velocity [60,64]) were probably not related to the effects of accuracy constraints or to a speed-accuracy trade-off, which are known to have impacts [65,66,67]. The main limit of the present work lies in the relatively high interaction efforts observed at the level of the arm interface. For extended applications, such interaction efforts could result in discomfort, skin issues and, eventually, rejection of the device by the user [55]. Furthermore, the effects of our interfaces would also need to be assessed during more complex movements to progress toward concrete applications. Nevertheless, it should be noted that shoulder and elbow sagittal plane movements are common types of movements in rehabilitation procedures [43,44,45,46], which are some of the main applications in active upper-limb exoskeletons.

## 5. Conclusions

Overall, it can be concluded that the proposed design, including passive rotations in the forearm interface, significantly improved the quality of the human-exoskeleton interaction when compared to common (i.e., fixed) human-exoskeleton interfaces. Such an interface design is a new compromise between the mechanical complexity and inertia of the device, as well as the interaction quality. Indeed, although this design did not allow for clearly reducing the interaction forces at the arm level, it implied limited additional inertia and was, overall, well-graded by the participants. Furthermore, it mainly introduces one large JM that can be efficiently identified following a simple procedure [50]. Finally, the proposed design seems to be suitable for clinical interventions or work tasks and for the burgeoning field of human movement studies involving exoskeletons [51,58,68]. In future works, the inclusion of passive degrees of freedom at the arm interface will be performed on the basis of the present study to further minimize unwanted interaction efforts while implementing a relatively simple design. The performances of this improved design will also need to be assessed with more complex controllers, for example, adaptive ones [69].

## Figures and Tables

**Figure 1 sensors-23-04122-f001:**
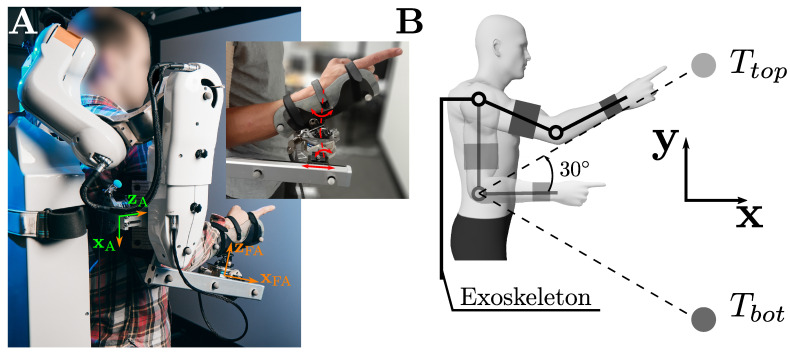
Illustration of the task. (**A**) Posture of a participant inside the exoskeleton. The axes ($x_{A}$ and $z_{A}$) of the arm force sensor are highlighted in green. The last axis $y_{A}$ of the arm force sensor was defined by $y_{A}=z_{A}\times x_{A}$. The same definitions were adopted for the forearm force sensor, with axes represented in orange. The passive rotations and translation are illustrated in red in the zoomed picture. (**B**) Schematic representation of the task and the target location. The reference position is shaded.

**Figure 2 sensors-23-04122-f002:**
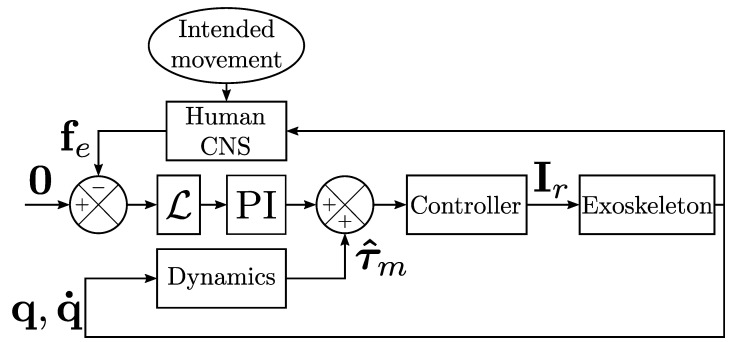
Control structure of the exoskeleton (see the description of Equation (Equation 1) for the definition of the terms).

**Figure 3 sensors-23-04122-f003:**
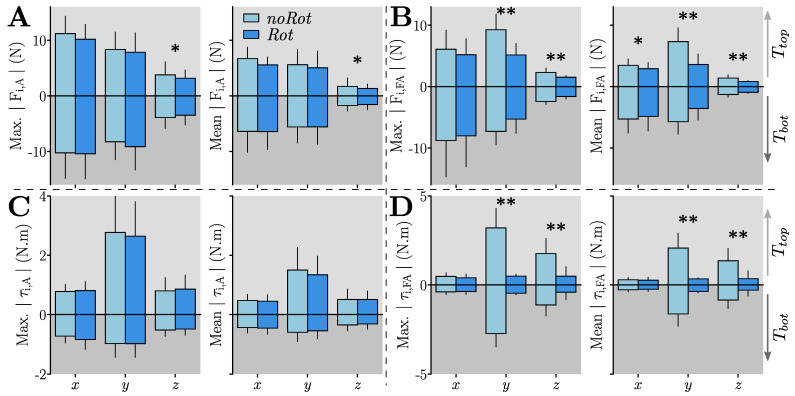
Maximum and averaged interaction effort components (i.e., $i\in \left\{x,y,z\right\}$) at the level of the arm (A) and forearm (FA) for both the *Rot* and *noRot* conditions. $T_{top}$ movements are represented with positive values and $T_{bot}$ movements are represented with negative values. Significant ANOVAs on the tested conditions are represented by “*” if p<0.05 and by “**” if $p<10^{-3}$. (**A**) Components of the interaction force at the arm. (**B**) Components of the interaction force at the forearm. (**C**) Components of the interaction torque at the arm. (**D**) Components of the interaction torque at the forearm.

**Figure 4 sensors-23-04122-f004:**
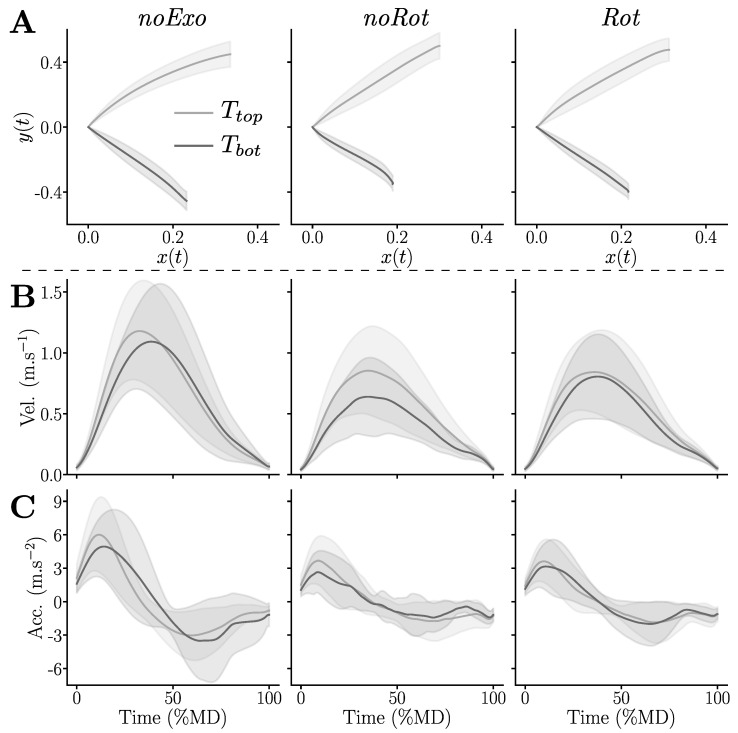
Average trajectories recorded in the three tested conditions for each of the two targets, $T_{top}$ and $T_{bot}$. (**A**) Consecutive positions of the index in the sagittal plane (x,y) (see Figure 1B). (**B**) Velocity profiles normalized by MD. (**C**) Acceleration profiles normalized by MD.

**Figure 5 sensors-23-04122-f005:**
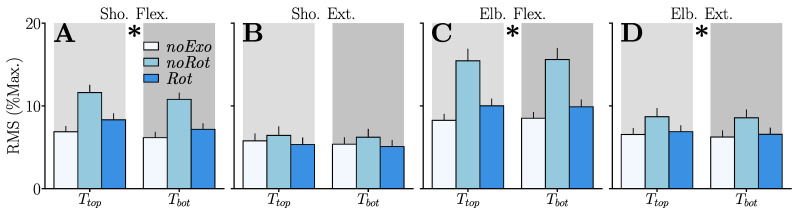
Average RMS of muscle groups for both targets and each condition. A main effect of the condition on the RMS is signaled by “*”. All significant ANOVAs returned p<0.05. (**A**) Shoulder flexors. (**B**) Shoulder extensors. (**C**) Elbow flexors. (**D**) Elbow extensors.

**Figure 6 sensors-23-04122-f006:**
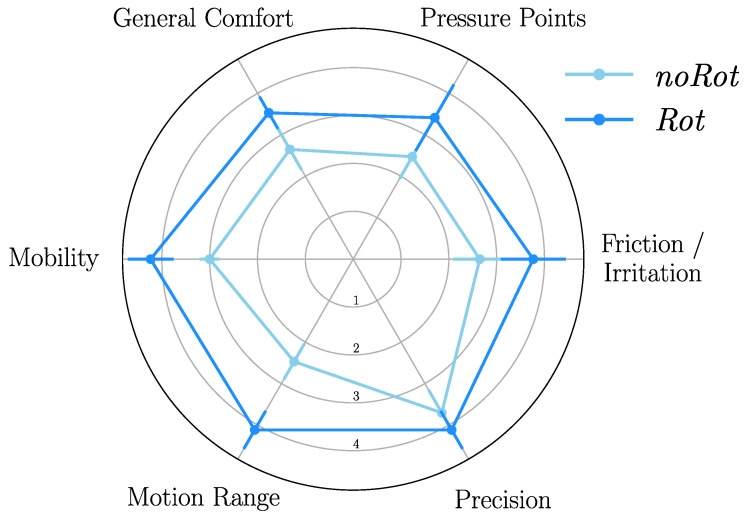
Average answers of the participants to the ergonomic feedback questionnaire.

**Table 1 sensors-23-04122-t001:** Summary of the computed descriptors of the kinematics for each condition. The ANOVA significance level associated with each parameter is given in the last column.

Parameter	Condition	$T_{\mathrm{top}}$	$T_{\mathrm{bot}}$	ANOVA
	*noExo*	1.22±0.34	1.16±0.37	
MD (s)	*noRot*	1.49±0.52	1.54±0.56	$p=3.10^{-5}$
	*Rot*	1.51±0.51	1.40±0.45	
	*noExo*	0.64±0.08	0.58±0.08	
Amp. (m)	*noRot*	0.63±0.11	0.50±0.07	p=0.006
	*Rot*	0.62±0.08	0.54±0.08	
	*noExo*	0.59±0.20	0.57±0.20	
aV ($m.s^{-1}$)	*noRot*	0.49±0.19	0.38±0.15	$p=8.10^{-8}$
	*Rot*	0.48±0.18	0.43±0.15	
	*noExo*	3.38±1.98	3.42±2.12	
mA ($m.s^{-2}$)	*noRot*	2.51±1.56	1.94±1.48	$p=3.10^{-6}$
	*Rot*	2.38±1.40	2.22±1.54	

## Data Availability

All data reported in this paper will be shared by the corresponding author upon request. Any additional information required to reanalyze the data reported in this paper is available from the corresponding author upon request. All data have been totally anonymized.

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
