# Peer review of "A Trade-Off between Complexity and Interaction Quality for Upper Limb Exoskeleton Interfaces"

_sensors, 2023, doi:10.3390/s23084122_

Round 1

Reviewer 1 Report

This manuscript analyzes a tradeoff between complexity and interaction quality for upper limb exoskeleton interfaces. This paper investigates the influence of two passive rotations in the forearm interface during sagittal plane reaching movements while keeping the arm interface unchanged.

  1. Introduction: The contribution of the manuscript was not defined. Several manuscripts reviews have reviewed the developments of exoskeleton systems: 

Anam K, Al-Jumaily AA. Active exoskeleton control systems: State of the art. Procedia Engineering. 2012 Jan 1;41:988-94.

Ferguson PW, Shen Y, Rosen J. Hand exoskeleton systems—Overview. Wearable Robotics. 2020 Jan 1:149-75.

Moreover, the kinematic complexity of the exoskeleton structures has also been analyzed:

Voilqué, A., Masood, J., Fauroux, J. C., Sabourin, L., & Guezet, O. (2019, March). Industrial exoskeleton technology: classification, structural analysis, and structural complexity indicator. In 2019 wearable robotics association conference (wearracon) (pp. 13-20). IEEE.

Yeem, S., Heo, J., Kim, H., & Kwon, Y. (2018). Technical analysis of exoskeleton robot. World Journal of Engineering and Technology7(1), 68-79

Panich, S. (2010). Kinematic analysis of exoskeleton suit for human arm. Journal of Computer Science6(11), 1272.

Muramatsu, Y., Kobayashi, H., Sato, Y., Jiaou, H., Hashimoto, T., & Kobayashi, H. (2011). Quantitative Performance Analysis of Exoskeleton Augmenting Devices-Muscle Suit-for Manual Worker. Int. J. Autom. Technol.5(4), 559-567.

Guo, S., Zhang, F., Wei, W., Guo, J., & Ge, W. (2013, May). Development of force analysis-based exoskeleton for the upper limb rehabilitation system. In 2013 ICME International Conference on Complex Medical Engineering (pp. 285-289). IEEE.

Yang, K., Jiang, Q. F., Wang, X. L., Chen, Y. W., & Ma, X. Y. (2018, September). Structural design and modal analysis of exoskeleton robot for rehabilitation of lower limb. In Journal of Physics: Conference Series (Vol. 1087, No. 6, p. 062004). IOP Publishing

Additionally, the control system of exoskeletons was also analyzed:

Jatsun, S., Malchikov, A., & Yatsun, A. (2020). Comparative analysis of the industrial exoskeleton control systems. In Proceedings of 14th International Conference on Electromechanics and Robotics “Zavalishin's Readings” ER (ZR) 2019, Kursk, Russia, 17-20 April 2019 (pp. 63-74). Springer Singapore.

Therefore, it is necessary to define the contribution of the present research study. 

2. Readability: It is recommended to check the writing to improve readability. For example, the expression "On the other hand" is repeated twice in the second paragraph of the introduction.

3. The ABLE active upper limb exoskeleton reference should be added (page 3).

4. Why was this specific evaluation task in section 2.2 considered? Could another evaluation task be considered in this analysis? 

5. the blocs of the control system of Fig. 2 should be explained. For example, what does the \mathcal{L} block before the PI controllers represent? \tau_r was not defined. More details should be included. It is not evident understand the relationship between this control system and the results of section 3. 

Reviewer 2 Report

A Tradeoff between Complexity and Interaction Quality for Upper Limb Exoskeletons Interfaces

This paper suggests that exoskeletons have great potential for assisting human movement during rehabilitation and preventing musculoskeletal disorders at work. However, the design of exoskeletons is limited by a fundamental contradiction that impacts their effectiveness, which is that increasing the interaction quality requires the inclusion of passive degrees of freedom in the design of human-exoskeleton interfaces. The inclusion of passive degrees of freedom increases the exoskeleton’s inertia and complexity, making control more complex and unwanted interaction efforts more likely. The paper proposes a compromise solution involving the addition of two passive rotations in the forearm interface during sagittal plane reaching movements, while keeping the arm interface unchanged. The paper investigates the benefits of such a design in terms of interaction efforts, kinematics, electromyographic signals, and subjective feedback from participants.

 Remarks:

1.      A flowchart of the methodology of the paper in developing the research should be included into the introduction to increase the clarity of the article.

2.      What are the metrics of the quality and performance of the robotic exoskeleton? And which are the impact of the proposed design solution on the overall comfort, usability, and safety of exoskeletons for users, especially during prolonged use.

3.      English language must be revised (ex. described and. Finally, etc.)

4.      Based on your research you should detail in the discussions:

a.      How can exoskeletons be designed to maximize interaction quality without increasing complexity and unwanted interaction efforts?

b.      and what are the criteria for determining the most suitable design parameters for rehabilitation applications.

5.      "Privacy and data integrity of the patients" are very important. This point should be discussed more in results or discussion.

6.      EMG data and results should be presented (better) in methods and results section.

7.      Improve conclusions. What does this paper add new to the literature?

8.      As future research you should explore the potential benefits of combining passive degrees of freedom with active control strategies, such as adaptive impedance control or machine learning approaches, to enhance the overall performance and adaptability of exoskeletons in various contexts.

Reviewer 3 Report

Comments to the Author

For the design of human-machine interaction in the exoskeleton rehabilitation robot system, a design method with four trade-offs in terms of interaction forces, dynamics, sEMG and subjective feedback is proposed to improve the complexity of the exoskeleton system and enhance the quality of human-machine interaction. However, before the article is accepted, I would like the authors to revise the following issues:

1. In this paper, the authors only consider the movement of the volunteer-worn rehabilitation robot in the sagittal plane, which is rather one-sided for the design of human-computer interaction, because the movement of human upper limbs is a kind of movement in three-dimensional space, and the offset of joints (especially the shoulder joint) in the movement in the sagittal plane is very small, so this method cannot fundamentally guide the design of human-computer interaction. It is suggested that the authors add the movement task of volunteers wearing the rehabilitation robot and study it accordingly.

2. The horizontal and vertical coordinates of Figures 3 and 4 in the article are not clearly expressed, and the contrast effect indicated in Figure 6 is not very obvious, so it is suggested to be revised.

3. In the paper, the authors compare the human-computer interaction with and without passive degrees of freedom, but the specific form and installation position of the passive degrees of freedom are not introduced in detail, so I hope the authors will supplement and improve them.

4. For research on human-robot interaction of upper limb exoskeleton robots, authors are advised to refer to the following articles:

       [1] Li, Jianfeng, et al. "Compatibility evaluation of a 4-DOF ergonomic exoskeleton for upper limb rehabilitation." Mechanism and Machine Theory 156 (2021): 104146.

       [2] Ning, Yuansheng, et al. "Design, optimization, and analysis of a human-machine compatibility upper extremity exoskeleton rehabilitation robot." Proceedings of the Institution of Mechanical Engineers, Part C: Journal of Mechanical Engineering Science (2022): 09544062221139988.

       [3] Jarrassé, Nathanaël, and Guillaume Morel. "Connecting a human limb to an exoskeleton." IEEE Transactions on Robotics 28.3 (2011): 697-709.

       [4] Ergin, Mehmet Alper, and Volkan Patoglu. "ASSISTON-SE: A self-aligning shoulder-elbow exoskeleton." 2012 IEEE international conference on robotics and automation. IEEE, 2012.

Round 2

Reviewer 1 Report

The authors included all the corrections. 

Reviewer 3 Report

This article has been fully revised, and it is recommended to publish it.